**Data Availability Statement:** All relevant data are within the manuscript.

**Funding:** This work is supported by the Natural Science Foundation of Zhejiang Province (CN) (by

# A new scheduling method based on sequential time windows developed to distribute first-aid medicine for emergency logistics following an earthquake

Jiaqi Fang[1], Hanping Hou📷[1]\*, Changxiang Lu📷[1], Haiyun Pang[2], Qingshan Deng[3], Yong Ye[4], Lingle Pan[5]\*

1 School of Economics and Management, Beijing Jiaotong University, Beijing, China, 2 School of Economics and Management, Zhejiang University of Science and Technology, Hangzhou, Zhejiang, China, 3 Department of Radiology, the First Affiliated Hospital of Wenzhou Medical University, Wenzhou, Zhejiang, China, 4 School of Public Health and Management, Wenzhou Medical University, Wenzhou, Zhejiang, China, 5 College of Emergency Management, Zhejiang College of Security Technology, Wenzhou, Zhejiang, China

\* hanpinghou@126.com (HH); panlynne@126.com (LP)

## Abstract

After an earthquake, affected areas have insufficient medicinal supplies, thereby necessitating substantial distribution of first-aid medicine from other supply centers. To make a proper distribution schedule, we considered the timing of supply and demand. In the present study, a "sequential time window" is used to describe the time to generate of supply and demand and the time of supply delivery. Then, considering the sequential time window, we proposed two multiobjective scheduling models with the consideration of demand uncertainty; two multiobjective stochastic programming models were also proposed to solve the scheduling models. Moreover, this paper describes a simulation that was performed based on a first-aid medicine distribution problem during a Wenchuan earthquake response. The simulation results show that the methodologies proposed in this paper provide effective schedules for the distribution of first-aid medicine. The developed distribution schedule enables some supplies in the former time windows to be used in latter time windows. This schedule increases the utility of limited stocks and avoids the risk that all the supplies are used in the short-term, leaving no supplies for long-term use.

## Introduction

An earthquake is a devastating natural disaster that can trigger secondary disasters, such as ground fracturing, earth splitting, landslides, debris flow, and tsunamis, thereby causing tremendous casualties and property losses due to the ensuing disaster chain. Major earthquakes have occurred frequently in history, such as the Kanto earthquake in Japan in 1923, the Tangshan earthquake in China in 1976, the Sumatra earthquake in Indonesia in 2004, the Pakistan earthquake in 2005, the Wenchuan earthquake in China in 2008, the Haiti earthquake in 2010,

YY, Grant No. LY20G010009 and by HYP, Grant No. Y17G030052), http://www.zjnsf.gov.cn/; National Key R&D Plan of China(by HPH, Grant No. 2016YFC0803207), https://service.most.gov.cn/; YY,JQF, Natural Science Foundation of China (Grant No. 71601146) http://www.nsfc.gov.cn/; Humanities and Social Sciences Foundation of Ministry of Education of China (by HYP, Grant No. 17YJC630109), https://www.sinoss.net/. The funders had no role in study design, data collection and analysis, decision to publish, or preparation of the manuscript.

**Competing interests:** The authors have declared that no competing interests exist.

and the east Japan earthquake in 2011. Norio et al. [1] believe that the main reason for the large-scale losses caused by Japan earthquakes is that the intensity of the earthquake and the resulting tsunami exceeded the local response capacity.

These catastrophes have caused tens of thousands, and potentially even hundreds of thousands, of deaths. The number of injured people following these earthquakes is shocking. In contrast to other disasters, earthquakes are characterized by sudden destructiveness that is difficult prevent. Therefore, the emergency response and rescue work after an earthquake are very important. The demand for food, medicine, water, tents, and other materials in an earthquake disaster area quickly and sharply increases. This demand is difficult to meet using locally reserved resources. According to announcements issued by local governments the shortage of emergency supplies was the greatest problem in the disaster relief and response processes following the Wenchuan, Yushu, Ya'an, Ludian, and Jiuzhaigou earthquakes.

The research of Zhang et al. [2] shows that emergency medical rescue is very important for post-earthquake rescue. The distribution of first-aid medicine following an earthquake is a continuous, multistage process. Prior studies have shown that the number of patients with medical conditions (e.g., wounded patients, patients with nontraumatic illness, and patients with internal disease, including acute respiratory tract infection, acute hemorrhagic enteritis, acute enteritis, and so on) surges shortly after an earthquake; subsequently the demand for emergency medicine increases, especially disinfectants and drugs used for-infections, anesthesia, and maintaining hemostasis. Even in developed countries, large-scale disasters inevitably lead to widespread drug shortages [3].

During the response process, the lives of injured patients are often seriously threatened if emergency medicines are of limited supply or their arrival to the disaster site is delayed. In addition, the intensities of the earthquake, secondary disaster chain, injuries of victims and other changes with time, may influence the actual time when demand for medical supplies surges. Thus, decision makers hope that the current distribution strategy can cope with future demand surges, supply delays, and other uncertainties. Some of the medicines available at an early stage can be reserved for subsequent stages, if necessary, to avoid the significant risk of drug unavailability. By using information from different time windows across the entire sequence, supply and demand between the early and late time windows can be adjusted and distributed and the ability of this first-aid medicine distribution strategy to attenuate future uncertainty can be enhanced.

This work focuses on two features: high time urgency and reduced need for transportation resources, which are particularly important. This article aimed to solve the problem of mismatch between supply and demand of medicines in earthquake disasters, so that emergency medicines can be distributed efficiently and quickly to reduce the losses caused by insufficient supply of medicines. The main contributions of this article are: the paper propose the definition of a time window sequence to divide the rescue stage scientifically, so that the rescue can be carried out orderly; this paper comprehensively considered the uncertainty of demand and other information, and suggested an earthquake first-aid medicine distribution mode used under the constraint of the time window sequence, developed an earthquake first-aid medicine distribution decision-making method, and then verified the feasibility and effectiveness of the decision-making strategy through a numerical simulation analysis.

The main contents of this paper are as follows: Firstly, this paper reviews the current research status and breakthrough in this field through literature review, and then proposes the earth quake first aid medicine distribution method under the constraint of a sequence of time windows. Through the simulation analysis of the medical supply in Wenchuan earthquake, the conclusion is drawn.

## Literature review

The distribution of emergency resources, such as first-aid medicine, is one of the main components of an operational emergency logistics system after a disaster. Relevant studies have been conducted on the application of random planning [4, 5], robust optimization [6], Bayesian analysis [7], spatiotemporal networks, and mixed-integer programming [8, 9]. These methods have been used to establish emergency resource distribution and scheduling models that include the uncertainty of emergency resource distribution, transport center location, distribution path selection, and other factors [10]; these models seek to meet the needs of disaster areas while considering both efficiency and fairness.

In the decision-making process regarding the distribution of resources following an earthquake, Feng [11] performed the following: grouped the affected areas by fuzzy clustering and material demand prioritization; estimated the emergency resource demand; established a weight-based material distribution model by considering the shortest time, the smallest loss, and the largest effect function; and added the minimum level of protection factor to the constraints to ensure the minimum demand for each affected area. This scholar's demand-based resource distribution model is quite enlightening. Although he emphasized the urgency of resource distribution after an earthquake through the objective function, he considered only the distribution of two kinds of life-sustaining emergency supplies. Moreover, with a cluster group study, Feng primarily considered the more obvious static indicators rather than the dynamic indicators, such as the death toll rise. Xia [12] presented the distribution of emergency materials based on demand analysis and made dynamic decisions on material distribution by establishing a demand estimation model based on multiple regression, a demand classification model based on a probability neural network, and an emergency material distribution model based on the needs and prioritization of disaster areas. Xia used genetic algorithms to calculate and analyze the demand estimation model. In contrast to Feng's fuzzy clustering method, Xia used the probabilistic neural network to classify the material demand, further optimizing the time urgency, but did not consider the correlation of various material needs and the reliability of decision-making schemes, risks, or other factors. Scholars have discerned that using demand is a reasonable way to realize the distribution of emergency materials in a natural disaster. Through the classification of demand and disaster areas, the demand can be accurately estimated, and the distribution weight can be obtained when the distribution decision is considered. Decision-making needs to be aligned as closely as possible to the specific situation at the earthquake site to optimize the distribution time, reduce the waste of resources, and solve the series of problems caused by a mismatch between supply and demand. Zhang [13] considers the problem of emergency resource allocation by evaluating primary and secondary disasters at the same time. In this paper, we propose a scenario tree based on conditional probability to define the relationship between primary and secondary generations of disasters and establish a multiobjective three-stage stochastic programming model for transportation time, transportation cost and unmet demand minimization.

Sun [14] proposed a dual-objective emergency logistics scheduling model, which includes transport time and transport cost, considering the uncertain traffic conditions and the actual road conditions. Mehrotra et al. [15] designed a stochastic optimization model for allocating and sharing key resources during a pandemic, and examined the distribution of ventilator inventory by the Federal Emergency Management Administration in different states in the United States during the COVID-19 pandemic using this model.

In contrast to more general emergency rescue responses, the shortage or delay of emergency medical supplies, particularly first-aid drugs (such as disinfectants, anti-infectives, anesthetics, hemostatics, and plasma) will result in untimely treatment of the disaster victims,

thereby leading to serious consequences. Biswas & Das [16] proposed a method based on fuzzy-analysis hierarchy process (Fuzzy-AHP). Research has shown that manpower shortage is the most important factor in the supply shortage of major epidemics. In an earthquake that occurred in Indonesia, a large number of victims died due to the shortage of medications. Many victims who survived both the earthquake and the tsunami could not be treated due to the lack of necessary medication. Eventually, these victims died of pneumonia or other infections due to the inhalation of large quantities of foreign bodies that were in the air. Many victims suffered worsening injuries due to untreated wounds; however, doctors were unable to treat them. Therefore, compared with those used as general emergency resources, first-aid medical resources for use in earthquake disasters must be characterized as priority resources that must be delivered with reduced transportation assets in an urgent manner. Currently, very little research has been done on this subject. The concept of a medical emergency was put forward very early in the United States. Thompson [17] proposed that the distribution of emergency medical and health resources, including medical equipment, medicine and medical aid, presents a problem. Ardekani et al. [18] analyzed emergency logistical issues following an earthquake. Fiedrich et al. [19] constructed a dynamic optimization decision-making model for emergency supply distribution after an earthquake. Najafi et al. [20] designed a dynamic model for earthquake emergency logistics and casualty transportation. Wang et al. [21] established a fuzzy dynamic LRP optimization model with time windows for a post-earthquake emergency logistics system. Guo [22] set targets to maximize time satisfaction and minimize system costs when building a network to meet the demand for drugs, thus improving the efficiency, reliability and quality of emergency medicine logistics. From the perspective of information management, experts have noted that the construction of an information system is the foundation of medical emergency logistics. Liu [23] proposed a genetic algorithm approach to minimize the unmet demand for medical supplies in all disaster areas and on the premise of minimizing the scheduling time, and constructed a dual-objective scheduling optimization model for emergency scheduling for minimizing the scheduling time and unmet demand. Minimizing the scheduling times is essential to match the supply and demand. This research emphasized urgency in logistics of distribution in an emergency. He [24] proposed a dynamic stochastic programming model for the distribution of medical supplies in the context of a large-scale infectious disease outbreak, studied the demand prediction and information sharing for medical supplies, and systematically solved the distribution requirements and strategies for emergency medicine. Using sudden cardiac death as an example, Wang [25] aimed to maximize the survival rate of sudden diseases by optimizing a multistage pharmaceutical logistics system and establishing a survival distribution model.

As a seismic event evolves, decision-makers should be concerned about when, where, how, and how much when allocating resources, along with considering how to generate the time-window sequence of demand. There is not a single independent time window in this scenario. Instead, multiple time windows occur in the form of a "sequence." The vehicle routing problem with time windows (VRPTW) is primarily used in the field of emergency logistics to assess the selection of routes and schedules for vehicles to distribute emergency resources using time windows as a constraint. Tuzkaya et al. [26] used the mixed integer programming method to establish the emergency logistics network planning model under the time window constraint. Fan [27] used the emergency search algorithm to establish the open vehicle routing problem with a time window constraint. Pan [28] defined and categorized VRPTW by inserting a heuristic algorithm, after which, VRPTW was modeled and calculated. Chen [29] proposed a multigoal nonlinear site-path model with a half-time window to minimize transportation costs, maximize the satisfaction rate of demand materials at the disaster site, and maximize the ability of vehicles to navigate to the demand site. Schiffer [30] incorporated time windows into the

routing of electric commercial vehicles (ECVs) to address the difficulty of their limited driving range; the distance traveled, number of available charging stations and total number of ECVs needed to cover the operations were all minimized. Gschwind et al. [31] proposed a branch-cut-and-price algorithm by using time window constraints in the field of emergency logistics. In addition, Zhu et al. [32] applied time window constraints to optimize emergency rescue paths. Adhikary [33] applies sensitivity analysis to illustrate how decision makers can set their target profits by using the distributes free newsboy model under fuzzy random demand by changing the mark-up value, discount rate and demand bias. Giri et al. [34] studied the coping strategies of the supply chain under random customer demand. In this decision-making model, the severity of wounded victims is transformed into a time window constraint to prioritize the prompt treatment of seriously wounded victims.

On the basis of the research described above, we combined the concepts of earthquake medical emergency logistics, resource allocation, time windows, etc., to propose a new scheduling method for the distribution of emergency medicine based on continuous time windows. Thus, we hope to provide theoretical significance and extend the existing knowledgebase for earthquake emergency logistics.

## Earthquake first-aid medicine distribution method under the constraint of a sequence of time windows

### Assumptions and symbols

Historically, the response efforts following earthquakes, such as those in the Wenchuan, Haiti, and East Japan, are long-term processes. Thus, based on the time that demand surges, the distribution process of first-aid medicine after an earthquake is divided into several time windows. A sequence is composed of multiple continuous time windows, as shown in Fig 1. Therefore, it is necessary to make decisions according to supply and demand to determine the distribution first-aid medicine within different time windows. Moreover, it is desirable to let the time window sequence have $n$ time windows, which are connected to one another and form an entire rescue period. Within this sequence, $m$ moments exist, which are the times when the first-aid medicine supply moment occurs. In addition, within a time window, there are $p$ demand sites and $q$ supply centers. The following assumptions are made:

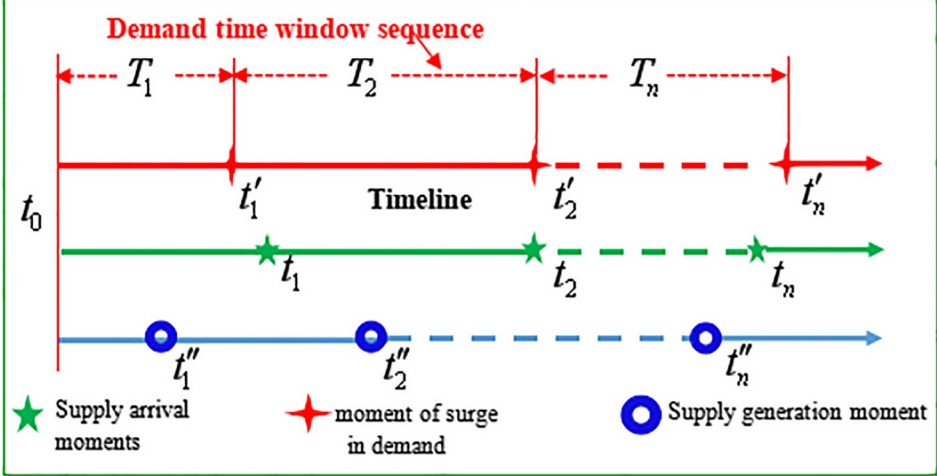

**Fig 1. Schematic diagram of the generation process for the time window sequence.**

*Assumption 1*: The decision is made after the earthquake outbreak, and coordinating the supply and demand distribution directly throughout the time window sequence is necessary to ensure minimum total loss during the period in which the entire time window series is located.

*Assumption 2*: The number and length of the time windows within a sequence are determined by the moment of demand surge, and decision makers can predict the moment of demand surge, the moment of supply generation, and so on.

*Assumption 3*: The amount of demand generated at each surge moment is uncertain, but the specific distribution can be determined from historical earthquake rescue information.

*Assumption 4*: The supply quantity at the supply generation moment can be predicted.

*Assumption 5*: As long as the supply is generated at the moment before the moment of demand surge in this time window, the generated supply can meet the demand of this time window; the supply generated at the supply generation moment cannot meet the demand generated at the moment of surge before this supply generation moment.

*Assumption 6*: Given the increased urgency and the need for reduced transportation resources, helicopters and drones are used for transportation. Therefore, the transport time in the model is directly expressed as that from the supply center to the demand site.

The symbols used in this paper are listed in Table 1.

**Table 1. Symbols.**

| Sets | | | |
|---|---|---|---|
| $\mathbf{T}$ | Time window set | $\mathbf{t}$ | Set of supply arrival moments |
| $\mathbf{t}'$ | Set of moments of surge in demand | $\mathbf{t}''$ | Set of supply generation moments |
| $\mathbf{J}$ | Demand site set | $\mathbf{K}$ | Supply center set |
| $\mathbf{I}$ | Set of natural numbers greater than 1, $i \epsilon \mathbf{I}$ | | |
| **Parameters** | | | |
| $j$ | demand site $j$, $j \epsilon \mathbf{J}$ | $k$ | supply center $k$, $k \epsilon \mathbf{K}$ |
| $T_i$ | time window $i$, $T_i \epsilon \mathbf{T}$ | $t'_i$ | moment of surge in demand $i$, $t'_i \in \mathbf{t}'$ |
| $t''_i$ | supply generation moment $i$, $t''_i \in \mathbf{t}''$ | $s_i^0$ | Supply amount generated at supply generation moment $i$ |
| $t_i$ | supply arrival moment $i$, $t_i \epsilon \mathbf{t}$ | | |
| $s_{1k}^0$ | The first time window, the maximum supply amount of the supply center $k$ | | |
| $t_{kj}^0$ | Transportation time from supply center $k$ to demand site $j$ | | |
| $\Delta t_i$ | The time difference value between the supply arrival moment and the moment of surge in demand in the time window $i$ | | |
| $t_{kj}^1$ | The first time window, the total transportation time from supply center $k$ to the demand site $j$ | | |
| **Stochastic variables** | | | |
| $\boldsymbol{\theta}$ | Demand amount | $\theta_j$ | demand amount at demand site $j$ |
| $\theta_j^1$ | Demand amount at the demand site $j$ in the first time window | | |
| $\Delta\theta_i$ | The difference value between the supply and demand in the time window $i$ | | |
| **Decision variables** | | | |
| $s_i$ | Supply amount at the time window $i$ | | |
| $s_{oi}$ | Supply amount from the supply generation moment $o$ to the time window $i$ | | |
| $s_{kj}^1$ | The first time window, the amount of distribution from supply center $k$ to demand site $j$ | | |
| **Function** | | | |
| $\Phi(\theta_i)$ | Probability density function of $\theta_i$ random variables | $\varphi(\theta_j^1)$ | Probability density function of $\theta_j^1$ random variables |

## Generation of the time window sequence

The demand-related time features were extracted from a summary of the historical evolution of earthquakes and practical rescue experiences and then used to generate a sequence of time windows for emergency medicine distribution; these features included earthquake suddenness, verticality, and renewal, in combination with the features inherent to the distribution of emergency medicine. Notable differences in the types of injuries and illnesses exist among the early, medium, and late stages of earthquake rescue. Consequently, it is necessary to classify the types of injuries that are prevalent in different periods. First, based on the large amount of collected data to be analyzed, cluster analysis was conducted according to when patients with various injuries were admitted and treated. The classification of injuries and division of "periods" were established using scatter and tree diagrams. Second, to define the primary injuries in each "period," specific time features were ascertained, such as when individuals were rescued, when wounded individuals were delivered, and when wounded individuals reached a medical treatment point. Finally, this information was combined with relevant historical data to analyze and predict the future demand time node, generating the time window for the distribution of various medicines. Fig 1 represents the signal of the generation process of the time window sequence: $t_0$ is the moment of earthquake outbreak; $t'$, $t'_2$, and $t'_n$ are the first, second, and $n$ demand surge moments, respectively; $T_1$, $T_2$, and $T_n$ are the first, second, and $n$ time windows, respectively; $t''_1$, $t''_2$, and $t''_n$ are the first, second, and supply generation moments $n$, respectively; $t_1$, $t_2$, and $t_n$ are the first, second, and $n$ supply arrival moments, respectively.

To describe the moment of surge in demand, the supply generation moment, and the supply arrival moment in the model, the values for each moment are defined as follows:

*Definition 1*: The surge in demand, supply generation, and supply arrival moments were defined as the duration of time between the moment that the earthquake began to the time when that particular activity occurred.

## Earthquake first-aid medicine distribution method under the constraint of a time window sequence

Due to the urgency of time, the lack of information and the surge in demand, formulating an effective drug distribution plan after the earthquake is a huge challenge [35]. For resources that need to be distributed in general emergency rescues, the specific time, location, and volume of distribution, along with the mode of transport, are considered. Moreover, relevant research has accounted for damage to roads caused by the earthquake, which affects the process of transportation and the selection of the mode of transportation, further complicating the decision-making model. However, given the priorities of using fewer transportation resources in an urgent manner for the distribution of earthquake first-aid medicine, helicopters and drones can be used directly when necessary. Therefore, for this situation, supply and demand were examined under the constraint of a time window sequence, which includes the specific time, quantity, and location of distribution. The definitions of earthquake first-aid medicine distribution method under the constraint of a time window sequence are as follows:

*Definition 2*: For each time window of the sequence, an optimal distribution must be determined according to its supply and demand information to meet the minimum demand for that time window, while also minimizing the utility loss.

The utility loss of the distribution of first-aid medicine indicates a failure to meet the minimum demand at the moment demand surges, thereby resulting in losses. Two aspects are included: (a) the functional relationship between the supply arrival moment and the moment

of surge in demand and (b) the functional relationship between supply and minimum demand. Therefore, the specific mathematical definitions are as follows:

*Definition 3*: The minimum demand for the time window $T_i$ was set as $\theta_i$, and the supply was set as $s_i$. Then, the utility loss of the earthquake first-aid medicine distribution in the time window $F_i$ can be expressed as follows:

$$F_i = f(\Delta\theta_i) \tag{1}$$

*Definition 4*: The moment of surge in demand at time window $T_i$, is set as $t'_i$, and the supply arrival moment is set as $t_i$. Then, the time loss of the earthquake first-aid medicine distribution $G_i$ can be expressed as follows:

$$G_i = g(\Delta t_i) \tag{2}$$

$\Delta\theta_i$ represents the difference value between supply and demand at the time window $i$, $T_i$ (3), and $\Delta t_i$ represents the time difference value between the supply arrival moment and the moment of surge in demand (4).

$$\Delta\theta_i = s_i - \theta_i \tag{3}$$

$$\Delta t_i = t_i - t'_i \tag{4}$$

## Decision method

### Distribution decision model throughout the time window sequence

The demand of each time window in the entire sequence is at maximum satisfaction; that is, a minimum distribution loss is present in each time window; thus, the establishment of the time window sequence of the earthquake first-aid medicine distribution decision model, M1, is as follows:

$$\min z_1 = \sum_{i\in\mathbf{I}} F = E\left[\sum_{i\in\mathbf{I}} f(s_i - \theta_i)\right] = \sum_{i\in\mathbf{I}} \int_{\mathbf{R}^+} f(s_i - \theta_i)\phi(\theta_i)d\theta_i \tag{5}$$

$$\min z_2 = \sum_{i\in\mathbf{I}} G_i = \sum_{i\in\mathbf{I}} g(t_i - t'_i) \tag{6}$$

to satisfy:

$$f(s_i - \theta_i) = \lambda \cdot (s_i - \theta_i)^2, \forall i \in \mathbf{I} \tag{7}$$

$$g(t_i - t') = 0, \ \ if \ (t_i - t') \le 0, \forall i \in \mathbf{I} \tag{8}$$

$$g(t_i - t') = \mu \cdot (t_i - t'), \ \ if \ (t_i - t') > 0, \forall i \in \mathbf{I} \tag{9}$$

$$s_i = \sum_{o\in\mathbf{o}_i} s_{oi}, \mathbf{o}_i = \{o|t''_o \le t'_i\} \tag{10}$$

$$\sum_{i\in\mathbf{I}} s_{oi} \le s_o^0, \forall o \in \mathbf{t}' \tag{11}$$

$$t_i = t''_i + t^0, \forall i \in \mathbf{I} \tag{12}$$

$$s_{oi}, s_i, t_i \in \mathbf{R}, \forall o \in \mathbf{t}', i \in \mathbf{I} \tag{13}$$

Eqs (5) and (6) are the target functions that minimize the earthquake first-aid medicine distribution utility loss and time loss, respectively. The difference between demand and supply is set to express the degree of matching between supply and demand. Considering that demand surges rapidly in the context of an earthquake, it is unrealistic to expect supply to match demand precisely. Whether supply is more or less than demand, the discrepancy will have an impact on the distribution. Thus, in expression (7) of the utility loss function of earthquake first-aid medicine distribution, the square value of the difference between demand and supply is calculated, with $\lambda$ as a penalty coefficient. However, the difference between the time that demand is satisfied and the moment that demand surges is used to express whether supply can arrive before demand generation. As long as supply is not delayed and can arrive before demand is generated, it will not impact the distribution effect. Thus, the function of utility loss in earthquake first-aid medicine distribution is expressed in (8) and (9), where $\mu$ is a penalty coefficient. Eq (10) shows that the supply of the time window $i$ can consist of all supply generation moments before the moment of demand surge in $i$. Eq (11) is used to create a supply constraint at the moment of supply for each supply, which cannot exceed the total amount of supply at the moment of subsequent surges in demand. Eq (12) is the calculation of the supply arrival moment, where $t^0$ is the basic supply transport time, and can be estimated by the specific time of transport. Eq (13) is the value constraint of the variable.

## Comprehensive distribution model at the first time window

By solving the distribution decision model in the entire time window sequence, the distribution amount of each time window can be obtained. At the same time, determination of the specific distribution scheme in the first time window is necessary to determine which supply center should distribute to each demand site, along with the volume of relief drugs to achieve the most efficient distribution. First, similar to Definition 3, the utility loss of the distribution of earthquake first-aid medicine within the first time window is defined. Second, the distribution efficiency is represented by the expression of transportation efficiency in the distribution scheme. Therefore, we use the transportation time to describe the distribution efficiency. As the time becomes shorter, the efficiency is improved. To maximize distribution efficiency, the comprehensive distribution model, M2, is built for the first time window as follows:

$$\min z_3 = E\left[\sum_{j \in \mathbf{J}} f(s_j^1 - \theta_j^1)\right] = \sum_{j \in \mathbf{J}} \int_{\mathbf{R}^+} f(s_j^1 - \theta_j^1)\varphi(\theta_j^1)d\theta_j^1 \tag{14}$$

$$\min z_4 = \sum_{k \in \mathbf{K}} \sum_{j \in \mathbf{J}} t_{kj}^1 \tag{15}$$

to satisfy:

$$f'(s_j^1 - \theta_j^1) = \lambda \cdot (s_j^1 - \theta_j^1)^2, \forall j \in \mathbf{J} \tag{16}$$

$$s_j^1 = \sum_{k \in \mathbf{K}} s_{kj}^1, \forall j \in \mathbf{J} \tag{17}$$

$$t_{kj}^1 = s_{kj}^1 \cdot t_{kj}^0, \forall k \in \mathbf{K}, j \in \mathbf{J} \tag{18}$$

$$\sum_{j \in \mathbf{J}} s_{kj}^1 \le s_{1k}^0, \forall k \in \mathbf{K} \tag{19}$$

$$s_{kj}^1, s_j^1 \in \mathbf{R}, \forall k \in \mathbf{K}, j \in \mathbf{J} \tag{20}$$

Eqs (14) and (15) are the target functions that respectively represent the earthquake first-aid medicine distribution utility loss and the time duration of first-aid medicine distribution. Eq (16) is the expression of the utility loss function of earthquake first-aid medicine distribution. Eq (17) is the total supply at demand site $j$. Eq (18) is the calculation expression for transportation time. Eq (19) is the supply constraint at the supply center $k$. Eq (20) is the value range of the variable.

## Solution method

The distribution decision model (M1), in the whole time window sequence, and the comprehensive distribution model (M2), at the first time window, are multi-objective stochastic programming models that are solved by objective programming. For M1, it may be desirable to first set the total expectation of earthquake first-aid medicine distribution utility loss of $F_1^0$ and the total expectation of earthquake first-aid medicine distribution time loss of $G_1^0$. Then, M1 can be converted to the objective programming model, M3, as follows:

$$\min z_5 = P'_1 \cdot d_1^+ + P'_2 \cdot d_2^+ \tag{21}$$

to satisfy:

$$\lambda \cdot \sum_{i \in \mathbf{I}} \int_{\mathbf{R}^+} (s_i - \theta_i)^2 \phi(\theta_i) d\theta_i + d_1^- - d_1^+ = F_1^0 \tag{22}$$

$$\sum_{i \in \mathbf{I}} g(t_i - t'_i) + d_2^- - d_2^+ = G_1^0 \tag{23}$$

It also satisfies Eqs (8)–(13). $P'_1$ and $P'_2$ are priority levels, and $d_1^+, d_1^-, d_2^-$, and $d_2^+$ are the positive and negative deviation variables of the two objective functions of M1.

For M2, it may be useful to set up the first time window of the earthquake first-aid medicine distribution utility loss and target value for the time spent as $F_2^0$ and $G_2^0$, respectively. Then, model M2 can be converted to the objective programming model, M4, as follows:

$$\min z_6 = P''_1 \cdot d_3^+ + P''_2 \cdot d_4^+ \tag{24}$$

to satisfy:

$$\lambda \cdot \sum_{j \in \mathbf{J}} \int_{\mathbf{R}^+} (s_j^1 - \theta_j^1)^2 \varphi(\theta_j^1) d\theta_j^1 + d_3^- - d_3^+ = F_2^0 \tag{25}$$

$$\sum_{k \in \mathbf{K}} \sum_{j \in \mathbf{J}} s_{kj}^1 \cdot t_{kj}^0 + d_4^- - d_4^+ = G_2^0 \tag{26}$$

It also satisfies Eqs (17), (19) and (20). $P''_1$ and $P''_2$ are priority levels, and $d_3^+$, $d_3^-$, $d_4^+$, and $d_4^-$ are the positive and negative deviation variables of the two objective functions of M2.

The solutions for M1 and M2 can be obtained by solving M3 and M4.

## Simulation analysis and discussion

### Simulation background

On Monday, May 12, 2008, at 14:28:04 (Beijing time; UTC-8), an earthquake occurred with a magnitude of 8.0 Ms and a moment magnitude of 8.3 Mw, according to data collected from the Earthquake Administration of the People's Republic of China. The U.S. Geological Survey, however, recorded the moment magnitude as 7.9 Mw and the intensity of the earthquake as reaching 11 degrees. The Wenchuan earthquake seriously damaged an area of more than 100,000 square kilometers, which included a total of 10 counties (cities). It severely affected an area covering 41 counties (cities), whereas the general disaster area comprised a total of 186 counties (cities). As a result, 69,227 people died, 374,643 were injured and 17,923 went missing. It was the most destructive earthquake since the founding of the People's Republic of China and the most serious earthquake following the Tangshan earthquake.

Rescue after the earthquake involves a series of material scheduling and distribution plans. Wex et al. [36] studied the allocation and scheduling of rescue units, and proposed a heuristic decision support model, which can quickly and effectively reduce casualties and economic losses in response phase. Zhou et al. [37] constructed a decomposition based multi-objective evolutionary algorithm (MOEA / D) model to solve the multi cycle dynamic emergency resource scheduling problem on the basis of non dominant sorting genetic algorithm II (NSGA-II). Ghasemi et al. [38] proposed improved multi-objective particle swarm optimization (MMOPSO), non dominant sorting genetic algorithm II (NSGA-II) and ε constraint algorithm to solve the uncertain multi-objective multi commodity multi cycle allocation model.

Previous studies focused on solving the distribution model, but ignored the urgency of time constraints. Our simulation posed Chengdu (CD), Deyang (DY) and Mianyang (MY) cities to serve as supply centers. The ten most extremely damaged areas, consisting of Wenchuan County (WC), Beichuan County (BBC), Mianzhu City (MZ), Shifang City (SF), Qingchuan County (QC), Maoxian County (MX), Anxian County (AX), Dujiangyan City (DJY), Pingwu County (PW), and Pengzhou City (PZ), were selected as the demand sites. A simulation of the supply and demand for distribution in 5 distinct time windows was performed. The logistics network of the 10 worst-hit areas in the Wenchuan earthquake are shown in Fig 2. Starting from the moment that the earthquake occurred, the duration of each time window, relating to either the surge in demand or the determination of supply generation, are shown in Table 2. The supply and demand forecast data for different supply centers and different demand sites at each time window are shown in Tables 3 and 4. In light of the small, yet highly important nature of the quantity of first-aid medicine, all rescue drugs were assumed to be transported by drones at a cruising speed of 170 km/h and a maximum capacity of 250 kg; the loading and unloading times were set to 30 minutes each. The transport, or supply-to-demand, time was determined to be the sum of the loading, unloading, and flight times, as shown in Table 5. If the demand at each demand site in each time window follows a normal distribution, then the demand at the demand site $j$ in the time window $T_i$ satisfies the distribution as follows:

$$\theta_j^{T_i} \sim N(\mu_j^{T_i}, (\sigma_j^{T_i})^2) \tag{27}$$

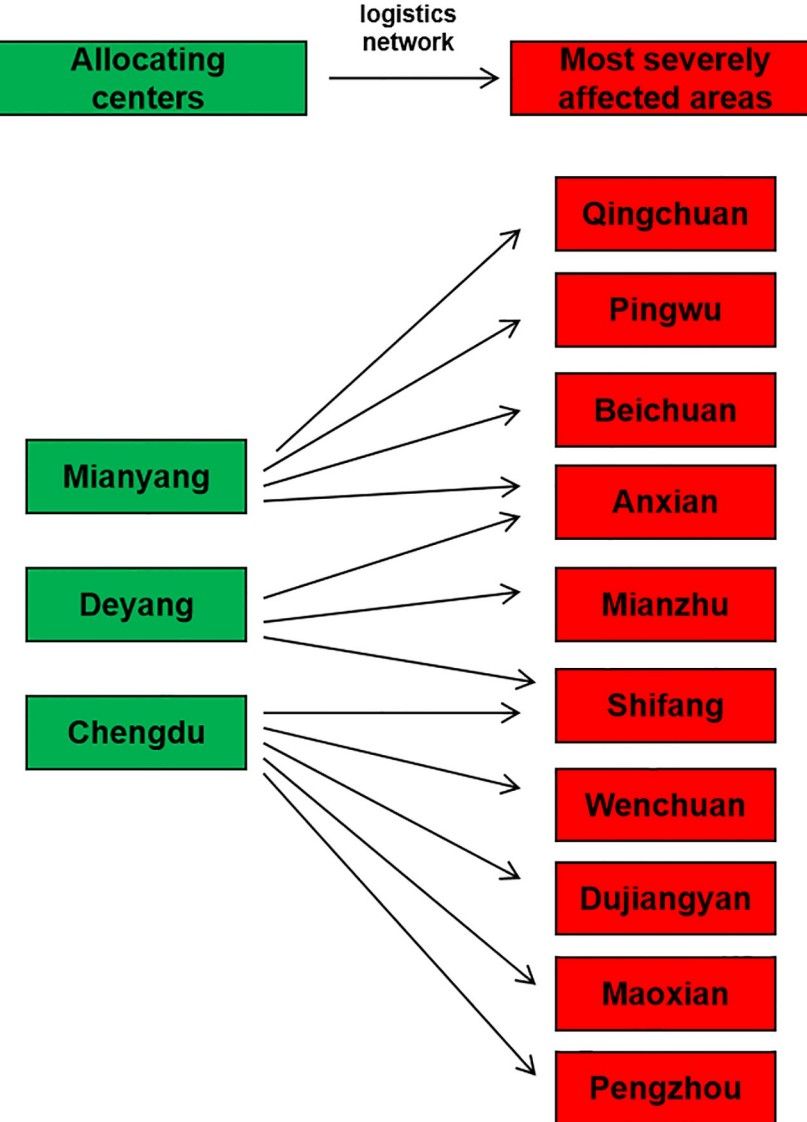

**Fig 2. Logistics network of the 10 worst-hit areas in the Wenchuan earthquake.**

If the requirements of different time windows and demand sites are independent, then the requirements of the time window $T_i$ satisfy the distribution as follows:

$$\theta^{T_i} \sim N(\sum_{j \in \mathbf{J}} \mu_j^{T_i}, \sum_{j \in \mathbf{J}} (\sigma_j^{T_i})^2) \tag{28}$$

The mean and variance for demand at each demand site from each time window can be predicted; the simulation generated a simple estimated demand based on the number of

**Table 2. Time window sequence.**

|  | Time window 1 | Time window 2 | Time window 3 | Time window 4 | Time window 5 |
|---|---|---|---|---|---|
| **Time window sequence of surge in demand** | 0–120 min | 121–130 min | 301–660 min | 661–960 min | 961–1200 min |
| **Time window sequence of supply generation** | 0–70 min | 71–260 min | 261–600 min | 601–720 min | 721–900 min |

**Table 3. Supply of each time window (unit: Box).**

|  | Time window 1 | Time window 2 | Time window 3 | Time window 4 | Time window 5 |
|---|---|---|---|---|---|
| Chengdu city | 10,568 | 8,023 | 10,882 | 6,317 | 8,576 |
| Deyang city | 5,765 | 4,012 | 5,441 | 3,159 | 4,288 |
| Mianyang city | 3,843 | 2,674 | 3,627 | 2,106 | 2,859 |

injuries at each affected site, as shown in Table 4. In addition, other parameters were set as follows: the basic transportation time of supply was $t^0 = 61.84$. The value of $\lambda$ was 1. The value of $\mu$ was 1; the utility loss and efficiency loss, represented by $F_1^0$, $F_2^0$, $G_1^0$, and $G_2^0$, were set to 0.

## Simulation results

Based on the data presented above, MATLAB and Lingo were used to solve M1 and M2 using the objective programming algorithm proposed in this paper, and the corresponding distribution results were obtained.

**Calculation result of M1.** The supply values from the five time windows are shown in Table 6. The total supply cannot meet the total expected demand; Supply window 1 produces a greater amount of supply than that of other supply windows. Supply window 1 not only satisfies the demand in window 1, but also supplies part of the demand in windows 2, 3, and 5. Although the supply from supply window 1 is greater than the expected demand of demand window 1, the expected demand of demand window 1 is not fully met in the specific distribution scheme due to the following reasons. The loss of the part of the out-of-stock of the demand window 1 is lower than that of out-of-stock of subsequent window. Thus, supply

**Table 4. The mean and variance of demand at each demand site in each time window (unit, box).**

|  | Number injured |  | Time window 1 | Time window 2 | Time window 3 | Time window 4 | Time window 5 |
|---|---|---|---|---|---|---|---|
| WC | 34583 | mean value | 3,458 | 2,767 | 3,804 | 2,075 | 3,112 |
|  |  | variance | 277 | 221 | 304 | 166 | 249 |
| BC | 9693 | mean value | 969 | 775 | 1,066 | 582 | 872 |
|  |  | variance | 78 | 62 | 85 | 47 | 70 |
| MZ | 36468 | mean value | 3,647 | 2,917 | 4,011 | 2,188 | 3,282 |
|  |  | variance | 292 | 233 | 321 | 175 | 263 |
| SF | 31990 | mean value | 3,199 | 2,559 | 3,519 | 1,919 | 2,879 |
|  |  | variance | 256 | 205 | 282 | 154 | 230 |
| QC | 15453 | mean value | 1,545 | 1,236 | 1,700 | 927 | 1,391 |
|  |  | variance | 124 | 99 | 136 | 74 | 111 |
| MX | 8183 | mean value | 818 | 655 | 900 | 491 | 736 |
|  |  | variance | 65 | 52 | 72 | 39 | 59 |
| AX | 13476 | mean value | 1,348 | 1,078 | 1,482 | 809 | 1,213 |
|  |  | variance | 108 | 86 | 119 | 65 | 97 |
| DJY | 4388 | mean value | 439 | 351 | 483 | 263 | 395 |
|  |  | variance | 35 | 28 | 39 | 21 | 32 |
| PW | 32145 | mean value | 3,215 | 2,572 | 3,536 | 1,929 | 2,893 |
|  |  | variance | 257 | 206 | 283 | 154 | 231 |
| PZ | 5770 | mean value | 577 | 462 | 635 | 346 | 519 |
|  |  | variance | 46 | 37 | 51 | 28 | 42 |
| Total | 192149 | mean value | 19,215 | 15,372 | 21,136 | 11,529 | 17,293 |
|  |  | variance | 1,537 | 1,230 | 1,691 | 922 | 1,383 |

**Table 5. Distance and transportation time from a supply center to the demand site (unit: Km, min).**

| Descriptions | | WC | BC | MZ | SF | QC | MX | AX | DJY | PW | PZ |
|---|---|---|---|---|---|---|---|---|---|---|---|
| CD | distance | 100 | 135 | 76 | 53 | 240 | 116 | 108 | 57 | 200 | 40 |
| | time | 65.29 | 77.65 | 56.82 | 48.71 | 114.71 | 70.94 | 68.12 | 50.12 | 100.59 | 44.12 |
| DY | distance | 85 | 78 | 30 | 22 | 180 | 80 | 48 | 76 | 143 | 45 |
| | time | 60.00 | 57.53 | 40.59 | 37.76 | 93.53 | 58.24 | 46.94 | 56.82 | 80.47 | 45.88 |
| MY | distance | 103 | 45 | 48 | 62 | 135 | 82 | 13 | 115 | 105 | 86 |
| | time | 66.35 | 45.88 | 46.94 | 51.88 | 77.65 | 58.94 | 34.59 | 70.59 | 67.06 | 60.35 |

**Table 6. Supply of each time window (unit: Box).**

| | Time window 1 | Time window 1 | Time window 1 | Time window 4 | Time window 5 | Total supply |
|---|---|---|---|---|---|---|
| Time window 1 | 18,734 | 182 | 705 | 0 | 555 | 20,176 |
| Time window 2 | 0 | 14,709 | 0 | 0 | 0 | 14,709 |
| Time window 3 | 0 | 0 | 19,950 | 0 | 0 | 19,950 |
| Time window 4 | 0 | 0 | 0 | 11,048 | 534 | 11,582 |
| Time window 5 | 0 | 0 | 0 | 0 | 15,723 | 15,723 |
| Total supply | 18,734 | 14,891 | 20,655 | 11,048 | 16,812 | 82,140 |

window 1 retains some supply to meet the demand of subsequent windows to minimize the total loss. Similarly, the supply of demand window 4 has some supply reserved to meet the demand of demand window 5.

**Calculation result of M2.** The supply values from three supply centers to 10 demand sites are shown in Table 7. The CD supply center provides anti-inflammatory drugs to WC, MZ, SF, MX, DJY, and PZ, along with six other demand sites, for a total of 9,128 boxes. A total of 1,440 boxes are not allocated and are, thus, reserved to use in the subsequent demand window. The DY supply center provides anti-inflammatory drugs to BC, MZ, AX, and PW in a total of 5,765 boxes. The GY supply center provides anti-inflammatory drugs to QC and PW in a total of 3,843 boxes. The specific distribution scheme is related to the time of delivery from each supply center to demand site, because the $P''_2$ level of distribution scheme is targeted as the desired level of distribution efficiency (minimum total transport time) after $P''_1$ by prioritizing the distribution quantity of each demand site.

## Discussion

As shown from the distribution in each time window (Table 6) and the first time window distribution scheme (Table 7), the demand satisfaction rates of each time window and each demand site are not the same (Figs 3 and 4). The figures show that, in general, the time windows and demand sites with greater demand have relatively higher demand satisfaction rates. This trend occurs because the goals of levels $P'_1$ and $P''_1$ of the distribution of utility loss are

**Table 7. The supply from three supply centers to 10 demand sites in the first time window.**

| To From | WC | BC | MZ | SF | QC | MX | AX | DJY | PW | PZ | Total supply |
|---|---|---|---|---|---|---|---|---|---|---|---|
| CD | 3,411 | 0 | 876 | 3,151 | 0 | 770 | 0 | 391 | 0 | 529 | 9,128 |
| DY | 0 | 921 | 2,723 | 0 | 0 | 0 | 1,300 | 0 | 821 | 0 | 5,765 |
| GY | 0 | 0 | 0 | 0 | 1,497 | 0 | 0 | 0 | 2,346 | 0 | 3,843 |
| Total supply | 3,411 | 921 | 3,599 | 3,151 | 1,497 | 770 | 1,300 | 391 | 3,167 | 529 | 18,736 |

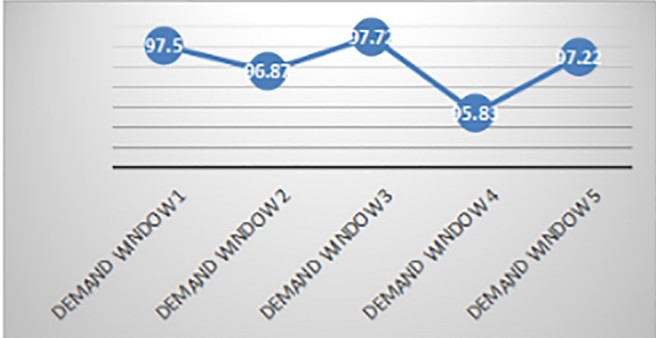

**Fig 3. Demand satisfaction rate of each demand window.**

minimized in the model established here and in the objective programming models M3 and M4. Thus, the demand satisfaction rates will be different. At the same time, the demand time window 1 reserves some anti-inflammatory drugs for demand windows 2, 3, and 5. The demand time window 4 also reserves some anti-inflammatory drugs for demand window 5. Therefore, the respective demand satisfaction rates of demand time windows 1 and 4 are only 97.5% and 95.83%, respectively (not fully satisfied). This result is consistent with the fact that the earthquake first-aid medicine distribution is a continuous, multistage process, and decision makers hope that the current distribution strategy can cope with future demand surges, supply delays, and other uncertainties. If necessary, some of the medicine at the current stage can be reserved for use in subsequent phases to avoid the significant risk of drug unavailability.

The simulation process of some studies is also similar to that of this paper. Yu [39] focuses on the performance of resource allocation, which is composed of efficiency, effectiveness, and equity, which correspond to economic cost, service quality and fairness, respectively. The work presented here differs from that of Yu in that human suffering is described as a deprivation cost in the utility measure, a nonlinear integer model is proposed, and an equivalent dynamic programming model is established to avoid the nonlinear term caused by the deprivation cost. Chen [40] proposed a two-layer planning model that accounts for problems in the distribution of disaster relief materials, such as road network destruction, high demand, material shortage and limited transportation capacity. The uncertainty and related factors

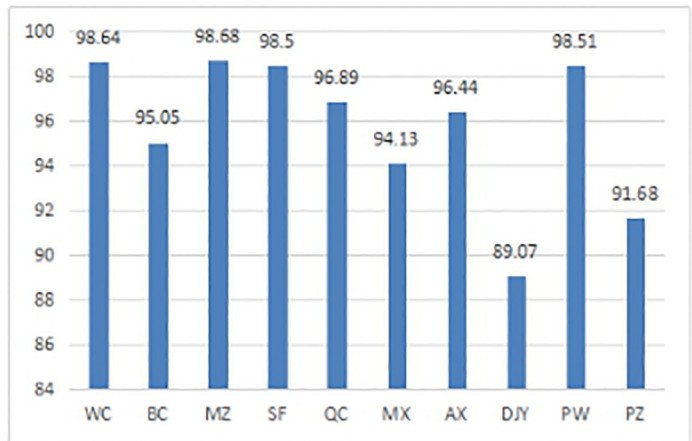

**Fig 4. Demand requirement rate of each demand site at the first time window.**

influencing the disaster relief process were effectively solved, and the feasibility and effectiveness of the model were verified by a simulation of the Wenchuan Earthquake that occurred in Western China in 2008. According to the characteristics and requirements of railway emergency resource scheduling, Tang and Sun [41] established a multi-objective optimization model aimed at minimizing the emergency resource scheduling time and the number of emergency rescue bases. The article uses Matlab software to design the decision-making process of emergency resource optimization dispatch, which makes emergency decision-making fast and scientific. At the same time, it conducts an empirical analysis on a railway bureau.

In the simulation process of this paper, considering the time urgency of post-disaster rescue, a "time series window" was proposed to divide the post-disaster rescue time, and a multi-objective optimization model was established to minimize the utility loss of drug distribution and the fastest rescue time. To verify the effectiveness of drug distribution scheme, we simulated the demand of anti-inflammatory drugs in 10 areas with the most severe earthquake disaster in Wenchuan.

## Conclusion

After an earthquake, an increased number of injured people in the disaster area leads to a sudden increase in the demand for anti-inflammatory drugs and other first-aid medicine; however, the locally reserved resources cannot meet the surge in demand. To better treat the injured population in the disaster area, the government needs to transport first-aid medicine from other areas. Given the need for fewer transportation resources and the time urgency for the distribution of earthquake first-aid medicine, the transportation conditions are less restrictive, but the specific distribution volume is very important. Therefore, considering the features of the distribution process, this paper establishes two multiobjective stochastic programming models for the distribution of first-aid medicine following an earthquake based on the time window sequence constraint and transforms it into two objective programming models to obtain a solution. We used the demand for anti-inflammatory drugs in the ten worst-hit areas of the Wenchuan earthquake as the input to perform a simulation and analysis of our developed model. The three prefecture-level cities of CD, DY, and MY were set up as supply centers. The simulation results revealed the following: (1) based on relevant data, the model presented here can provide an effective solution for the distribution of first-aid medicine and can ensure that the distribution utility loss is minimized, and (2) to maximize the effectiveness of limited first-aid medicine, decision makers must consider the demand and supply across the time window sequence. When necessary, a portion of the drugs supplied in early time windows should be reserved for use in subsequent demand windows. In the future research, different distribution strategies can be used to expand drug distribution methods in earthquake relief. In addition, scholars can also consider using Bayesian theory or other information updating theories to dynamically generate decisions according to the actual seismic situation changes. The uncertain factors considered in this paper are limited, and other scholars can also consider other uncertain factors according to the actual situation of the earthquake, so as to improve the accuracy of demand analysis.

## Author Contributions

**Data curation:** Qingshan Deng, Lingle Pan.

**Investigation:** Haiyun Pang.

**Writing – original draft:** Jiaqi Fang.

**Writing – review & editing:** Hanping Hou, Changxiang Lu, Yong Ye.

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
