## [Decision Letter · Decision Letter 0]

3 Jan 2021

PONE-D-20-25913

A new scheduling method based on sequential time windows developed to distribute first-aid medicine for emergency logistics following an earthquake

PLOS ONE

Dear Dr. Hou,

Thank you for submitting your manuscript to PLOS ONE. After careful consideration, we feel that it has merit but does not fully meet PLOS ONE’s publication criteria as it currently stands. Therefore, we invite you to submit a revised version of the manuscript that addresses the points raised during the review process.

We look forward to receiving your revised manuscript.

Kind regards,

Dragan Pamucar

Academic Editor

PLOS ONE

Journal Requirements:

2.We note that [Figure(s) 2] in your submission contain map images which may be copyrighted. All PLOS content is published under the Creative Commons Attribution License (CC BY 4.0), which means that the manuscript, images, and Supporting Information files will be freely available online, and any third party is permitted to access, download, copy, distribute, and use these materials in any way, even commercially, with proper attribution. For these reasons, we cannot publish previously copyrighted maps or satellite images created using proprietary data, such as Google software (Google Maps, Street View, and Earth). For more information, see our copyright guidelines: http://journals.plos.org/plosone/s/licenses-and-copyright.

1.    You may seek permission from the original copyright holder of Figure(s) [2] to publish the content specifically under the CC BY 4.0 license. 

Reviewers' comments:

Reviewer's Responses to Questions

**Comments to the Author**

1. Is the manuscript technically sound, and do the data support the conclusions?

Reviewer #1: Yes

Reviewer #2: Yes

2. Has the statistical analysis been performed appropriately and rigorously? 

Reviewer #1: Yes

Reviewer #2: Yes

3. Have the authors made all data underlying the findings in their manuscript fully available?

Reviewer #1: Yes

Reviewer #2: Yes

4. Is the manuscript presented in an intelligible fashion and written in standard English?

Reviewer #1: Yes

Reviewer #2: Yes

5. Review Comments to the Author

Reviewer #1: Paper is good but needs to be improved. In each chapter, in addition to the summary, results and conclusion, there should be references. References are placed in the paper only in the Literature review chapter. It is necessary to add references to other chapters, in order to improve the model you used in the literature paper. Insert a chapter on discussions, because you have a lot of simulations, explain them, and connect them with the previous papers.

Reviewer #2: The paper has potential for publication, but should be in some parts improved. Generally, paper deserves attention because deals with very important topic. Also, applied methodology is good.

Please improve your paper adopting the following suggestions:

- Introduction section should be more clear and precise. Please write clear aims and contributions.

- As the last paragraph in introduction please add short structure of the whole paper.

- Please consider and cite the following new references:

Biswas, T. K., & Das, M. C. (2020). Selection of the barriers of supply chain management in Indian manufacturing sectors due to COVID-19 impacts. Operational Research in Engineering Sciences: Theory and Applications, 3(3), 1-12.

Adhikary, K., Roy, J., & Kar, S. (2019). Newsboy problem with birandom demand. Decision Making: Applications in Management and Engineering, 2(1), 1-12.

Giri, B. C., & Dey, S. (2020). Game theoretic models for a closed-loop supply chain with stochastic demand and backup supplier under dual channel recycling. Decision Making: Applications in Management and Engineering, 3(1), 108-125.

Write guidelines for future research in conclusion.

6. PLOS authors have the option to publish the peer review history of their article (what does this mean?). If published, this will include your full peer review and any attached files.

Reviewer #1: No

Reviewer #2: No

---

## [Author Response · Author response to Decision Letter 0]

2 Feb 2021

(1)Reviewer #1 Comments: Paper is good but needs to be improved. In each chapter, in addition to the summary, results and conclusion, there should be references. References are placed in the paper only in the Literature review chapter. It is necessary to add references to other chapters, in order to improve the model you used in the literature paper. Insert a chapter on discussions, because you have a lot of simulations, explain them, and connect them with the previous papers.

Author response: In each chapter, I have added some literature related to the content of my article, and these literature can help to better understand my article. I inserted a chapter on simulation and related them to the previous paper, and I also added some content to help readers better understand the simulation process.

(2)Reviewer #2 Comments:The paper has potential for publication, but should be in some parts improved. Generally, paper deserves attention because deals with very important topic. Also, applied methodology is good.

Please improve your paper adopting the following suggestions:

- Introduction section should be more clear and precise. Please write clear aims and contributions.

- As the last paragraph in introduction please add short structure of the whole paper.

- Please consider and cite the following new references:

Biswas, T. K., & Das, M. C. (2020). Selection of the barriers of supply chain management in Indian manufacturing sectors due to COVID-19 impacts. Operational Research in Engineering Sciences: Theory and Applications, 3(3), 1-12.

Adhikary, K., Roy, J., & Kar, S. (2019). Newsboy problem with birandom demand. Decision Making: Applications in Management and Engineering, 2(1), 1-12.

Giri, B. C., & Dey, S. (2020). Game theoretic models for a closed-loop supply chain with stochastic demand and backup supplier under dual channel recycling. Decision Making: Applications in Management and Engineering, 3(1), 108-125.

Write guidelines for future research in conclusion.

Author response: I revised the introduction to make it more clear and precise, and wrote contributions and and aims to help the reader better understand. I added short structure of the whole paper in the last paragraph in introduction. After reading the recommended references, I thought it is helpful to cite these references in my article for readers. At last, I wrote guidelines for future research in conclusion.

---

## [Decision Letter · Decision Letter 1]

10 Feb 2021

A new scheduling method based on sequential time windows developed to distribute first-aid medicine for emergency logistics following an earthquake

PONE-D-20-25913R1

Dear Dr. Hou,

We’re pleased to inform you that your manuscript has been judged scientifically suitable for publication and will be formally accepted for publication once it meets all outstanding technical requirements.

Kind regards,

Dragan Pamucar

Academic Editor

PLOS ONE

Additional Editor Comments (optional):

Reviewers' comments:

Reviewer's Responses to Questions

**Comments to the Author**

1. If the authors have adequately addressed your comments raised in a previous round of review and you feel that this manuscript is now acceptable for publication, you may indicate that here to bypass the “Comments to the Author” section, enter your conflict of interest statement in the “Confidential to Editor” section, and submit your "Accept" recommendation.

Reviewer #1: All comments have been addressed

Reviewer #2: All comments have been addressed

2. Is the manuscript technically sound, and do the data support the conclusions?

Reviewer #1: Yes

Reviewer #2: Yes

3. Has the statistical analysis been performed appropriately and rigorously? 

Reviewer #1: Yes

Reviewer #2: Yes

4. Have the authors made all data underlying the findings in their manuscript fully available?

Reviewer #1: Yes

Reviewer #2: (No Response)

5. Is the manuscript presented in an intelligible fashion and written in standard English?

Reviewer #1: Yes

Reviewer #2: (No Response)

6. Review Comments to the Author

Reviewer #1: Greeitngs. The author followed all the suggestions and corrected the paper accordingly. Paper is better now and can be published.

Reviewer #2: The paper has potential for publication, but should be in some parts improved. Generally, paper deserves attention because deals with very important topic. Also, applied methodology is good.

Please improve your paper adopting the following suggestions:

- Introduction section should be more clear and precise. Please write clear aims and contributions. DONE

- As the last paragraph in introduction please add short structure of the whole paper. DONE

- Please consider and cite the following new references:

Biswas, T. K., & Das, M. C. (2020). Selection of the barriers of supply chain management in Indian manufacturing sectors due to COVID-19 impacts. Operational Research in Engineering Sciences: Theory and Applications, 3(3), 1-12.

Adhikary, K., Roy, J., & Kar, S. (2019). Newsboy problem with birandom demand. Decision Making: Applications in Management and Engineering, 2(1), 1-12.

Giri, B. C., & Dey, S. (2020). Game theoretic models for a closed-loop supply chain with stochastic demand and backup supplier under dual channel recycling. Decision Making: Applications in Management and Engineering, 3(1), 108-125. DONE

Write guidelines for future research in conclusion. DONE

Now the paper can be accepted.

7. PLOS authors have the option to publish the peer review history of their article (what does this mean?). If published, this will include your full peer review and any attached files.

Reviewer #1: No

Reviewer #2: No

---

## [Editor Report · Acceptance letter]

12 Feb 2021

PONE-D-20-25913R1 

A new scheduling method based on sequential time windows developed to distribute first-aid medicine for emergency logistics following an earthquake 

Dear Dr. Hou:

I'm pleased to inform you that your manuscript has been deemed suitable for publication in PLOS ONE. Congratulations! Your manuscript is now with our production department. 

Kind regards, 

on behalf of

Dr. Dragan Pamucar 

Academic Editor

PLOS ONE